# Exploring the relation between Interleukin-6 and high-sensitive cardiac troponin T in asymptomatic hemodialysis patient: A cross-sectional study

Leen Ibrahim[1][☉], Katreen Yasin[1][☉], Leen Abbas[1][☉], Yahya Ismael[1], Ahmed Mousa[2], Mohammad Alkarajeh[3], Zakaria Hamdan[1,3]*, Zaher Nazzal [1]*

1 Faculty of Medicine and Health Science, Department of Medicine, An-Najah National University, Nablus, Palestine, 2 Faculty of Medicine and Health Sciences, Biomedical and Sciences Labs, An-Najah National University, Nablus, Palestine, 3 Faculty of Medicine and Health Sciences, Department of Nephrology, An-Najah National University Hospital, Nablus, Palestine

☉ These authors contributed equally to this work.
* znazzal@najah.edu (ZN); z.hamdan@najah.edu (ZH)

**Data Availability Statement:** All relevant data are within the manuscript and its Supporting Information files.

## Abstract

### Background

High-sensitive cardiac troponin T (h-cTnT), which serves as a marker for myocardial damage, has also been linked to adverse outcomes in asymptomatic hemodialysis patients. This study aims to explore the correlation between interleukin-6 (IL-6) and h-cTnT in asymptomatic hemodialysis patients to unravel the relationship between inflammation and cardiovascular risk.

### Methods

A cross-sectional study involving 81 patients was conducted from November 2022 to March 2023 at An-Najah National University Hospital in Palestine. We gathered clinical data, including comorbidities, and obtained blood samples for measuring IL-6 and h-cTnT levels. We performed statistical analyses, including correlation tests and linear regression, to assess the associations between these variables.

### Results

The study revealed a notable increase in both h-cTnT and IL-6 levels, and a significant correlation between the two (rho = 0.463, P<0.001) in asymptomatic hemodialysis patients. Likewise, h-cTnT levels displayed positive correlations with age (rho = 0.519, P<0.001) and negative correlations with albumin (rho = -0.297, p = 0.007) and transferrin saturation (rho = -0.227, P = 0.042). IL-6 levels exhibited correlations with age (rho = 0.422, P<0.001), albumin (rho = -0.389, P<0.001), iron (rho = -0.382, P<0.001), and transferrin saturation (rho = -0.362, P = 0.001). Notably, higher h-cTnT levels were associated with diabetes, hypertension, a history of coronary artery disease, cerebrovascular accidents, older age, and male gender.

**Funding:** The author(s) received no specific funding for this work.

**Competing interests:** The authors have declared that no competing interests exist.

## Conclusion

This study underscores the significant association between the inflammatory marker IL-6 and h-cTnT in asymptomatic hemodialysis patients, suggesting that inflammation may play an essential role in the elevation of h-cTnT levels. This association may have implications for predicting cardiovascular events and guiding interventions to reduce cardiovascular disease morbidity and mortality in hemodialysis patients.

## Introduction

Chronic kidney disease (CKD) is a prevalent health condition affecting approximately 700 million people worldwide. The last stage of CKD is defined as an irreversible decrease in kidney function, where it refers to an estimated glomerular filtration rate (eGFR) of less than 15 ml per minute per 1.73 $m^2$ body surface area or those requiring dialysis [1]. Hemodialysis (HD) and peritoneal dialysis are currently used for end-stage renal disease (ESRD) treatment. These patients are at higher risk of developing cardiovascular diseases (CVD), resulting in higher mortality rates than in the general population [2].

Chronic inflammation is associated with CKD, even with higher prevalence in long-term dialysis patients [3]. Blood inflammatory markers like C-reactive protein and interleukin-6 (IL-6) were elevated in these patients [4]. Although the cause is poorly understood, multiple unique factors are associated with this process in long-term HD. Dialysis membranes, central venous catheters, oxidative stress, hypoxia, fluid overload, and immunological dysfunction are a few examples of this [5]. Inflammatory and oxidative stress markers increase during a single HD session, mimicking changes during acute immune activation.

The decline of renal function is associated with inadequate removal of uremic toxins that should be excreted by the kidney. As these substances are biologically active, they are both the cause and consequence of CKD. Uremic toxins have been shown to play a vital role in the onset of oxidative stress and development of inflammatory state [6]. It's found that a rise of inflammatory markers like IL-6 levels during a single HD session is associated with a higher mortality among HD patients, indicating that single dialysis session acutely increase the level of IL-6 [7]. This implies the presence of an intradialytic inflammatory response that affects survival in HD patients.

IL-6 is an inflammatory mediator that is elevated in patients with chronic HD [8]. It is also associated with coronary artery calcification [9]. The elevation of IL-6 level is associated with cardiovascular events and is a predictor of mortality in dialysis patients [10]. It is a better predictor of cardiovascular mortality among HD than other inflammatory markers like c-reactive protein [11].

It is well-recognized that inflammation is a significant factor in atherosclerosis pathogenesis and a potent risk factor for CVD in the general population. In individuals with ESRD, CVD continues to be the major cause of mortality and morbidity [12]. It is widely established that oxidative stress, endothelial dysfunction, and inflammation in the artery's wall all contribute to the development of atherosclerosis, which is an inflammatory process [13]. Although HD has been the most available method for the treatment of ESRD, it appears to have side effects on the cardiovascular system via increasing myocardial stress in an already compromised system. More than 50% of HD patients end up with CVD and death [14]. This is due to multiple cardiovascular changes associated with renal dysfunction, including fluid overload, uremic toxins accumulation, secondary hyperparathyroidism, anemia, and hyperlipidemia. With

more prolonged dialysis treatment, oxidative stress has also gradually increased, which decreases nitric oxide availability, contributing to endothelial dysfunction [15]. These alter the permeability of the blood vessels, allowing LDL cholesterol to enter the intima and trigger inflammatory cascades that lead to the development of atherosclerotic plaques. Myocardial infarction can occur due to the rupture of atherosclerotic plaque [16]. Among dialysis patients, 40% to 50% of deaths are caused by CVD, and 10% to 20% by acute myocardial infarction [17].

Troponin complex comprises three regulatory proteins (C, I, T) found in cardiac and skeletal muscles. High-sensitive cardiac troponin T (H-cTnT) is a marker of acute myocardial damage and a predictor of adverse outcomes in patients with unstable coronary disease [18]. Both troponin I and h-cTnT are the current markers of choice for detecting acute myocardial infarction. However, h-cTnT is preferred in detecting subclinical myocardial injury in asymptomatic patients [19]. Moreover, it predicts all-cause and cardiovascular mortality in HD patients [20]. It has been demonstrated that uremic toxins have a detrimental effect on cells involved in the functioning of myocardium and vessels, including smooth muscle cells, endothelial cells, and platelets leucocytes, thus affecting the cardiovascular system [6]. Some studies reporting increase in hs-cTnT levels during dialysis suggesting that a single HD session acutely increase the value of hs-cTnT during dialysis and this increase was only due to dialysis itself not due to CKD [21].

In people with unstable angina pectoris, an elevation in IL-6 and other inflammatory markers has been reported [22]. According to previous studies, its well-known that both of IL6 and h-cTnT are persistently elevated in patients with CKD [6,20]. However, the association between IL-6 and h-cTnT in asymptomatic HD patients remains unclear. Therefore, the present study aims to investigate a possible correlation between IL-6 that results from chronic inflammation and h-cTnT in asymptomatic HD patients. This will help early CVD detection and decrease mortality and morbidity among them.

## Methodology

### Study design and population

This cross-sectional study was conducted between November 2022 to March 2023 at the HD Center of An-Najah National University Hospital (NNUH), Palestine. NNUH, a teaching tertiary referral hospital has the largest dialysis center in the area. Eighty-one HD patients participated in this study. We included patients who were over the age of 18 and had been on HD for more than six months. Those with recent onset (less than two weeks) of symptoms of the acute coronary syndrome (chest pain radiating to jaw, neck, shoulders, or left arm) or with a known history of chronic inflammatory disease (rheumatoid arthritis, systemic lupus erythematosus, ankylosing spondylitis, psoriasis, and Crohn's disease), or those with underlying infection, and who are not reliable due to medical or psychological conditions were excluded from the study.

We calculated the sample size using the EpiInfo software's "Sample Size for Comparing Two Means" method, where the predicted h-cTnT in the IL6 high group was 57±10.2 and in the IL6 low group was 50±10.4. The confidence interval (2-sided) was 95%, and the power was 80%. This resulted in a sample size of 76. Participants were selected conveniently; 81 patients agreed to participate and met the criteria.

### Variables and measurement tool

Demographic and clinical data were collected from the participant's medical records. This includes age, gender, body mass index (BMI), duration of HD, type of vascular access, smoking habit, chronic respiratory disease, history of malignancies, diabetes, hypertension, previous

history of ischemic heart disease (IHD), cerebrovascular accident (CVA), and peripheral arterial disease (PAD). Patients were divided into groups according to the presence or absence of these comorbidities. We reviewed previous patients' medical records, looking for patients who underwent percutaneous cardiac catheterization to investigate if they had ectatic and \ or stenotic vessels.

Two blood samples were drawn in two tubes (EDTA tube and lithium heparin tube) for 81 patients. Samples were assayed for IL-6 and h-cTnT levels. Blood samples were centrifuged and stored at -20 c for one week until the samples were collected. The h-cTnT was analyzed by fifth-generation electroluminescence immunoassay using Cobas e 602(Roche Diagnostics). The detection limit was three ng/L, and the coefficients of variation below 10% were 11 ng/L. The cutoff at the 99th percentile was 14 ng/L.

Plasma IL-6 levels were measured using Diaclone human IL-6 enzyme-linked immunosorbent assay (ELISA) set. The sensitivity or minimum detectable limit for IL-6 was 2pg/mL, and the intraassay coefficient of variation was 3.6%. The normal value of IL-6 is below 7 pg/ml. Patients with high troponin levels above the upper reference limit of the average population (14ng/L) were assessed clinically if they had any signs or symptoms of acute coronary syndrome. Additionally, serum levels of albumin, calcium, iron, phosphate, potassium, sodium, hemoglobin, platelet count, WBC count, parathyroid hormone, and bicarbonate were measured at An-Najah National Hospital Laboratory with methods used in clinical routine.

**Table 1. Sociodemographic and clinical characteristics of the participants.**

|  | Frequency (%) | Median [IQR] |
|---|---|---|
| **Gender** |  |  |
| Male | 49 (60.5%) |  |
| Female | 32 (39.5%) |  |
| **Age** |  | 58 (20–83) |
| **BMI** |  | 27.7 (15.6–50.8) |
| Underweight | 2 (2.5%) |  |
| Normal | 25 (30.9%) |  |
| Overweight | 29 (35.8%) |  |
| Obese | 25 (30.9%) |  |
| **Smoker** |  |  |
| Yes | 39 (48.1%) |  |
| No | 42 (51.9%) |  |
| **Comorbidities** |  |  |
| Diabetes | 41 (50.6%) |  |
| Hypertension | 61 (75.3%) |  |
| IHD | 40 (49.4%) |  |
| CVA | 13 (16.0%) |  |
| PAD | 15 (18.5%) |  |
| **Duration of dialysis** (month) |  | 48 (7–240) |
| <18 months | 17(21.0%) |  |
| ≥18 months | 64(79.0%) |  |
| **Vascular Access** |  |  |
| Fistula | 70 (86.4%) |  |
| Catheter | 11 (13.6%) |  |

IQR interquartile range, BMI body mass index, IHD ischemic heart disease, CVA cerebrovascular accident, PDA peripheral arterial disease.

## Statistical analysis

Statistical Package for Social Sciences (SPSS) version 28 software was used for data entry and analysis. Descriptive analysis was used to describe the characteristics of participants using frequencies and percentages for categorical variables, mean ± standard deviation (SD) or medians, and interquartile ranges for continuous data. The normality of the data was checked using the Kolmogorov—Smirnov test. The h-cTnT and IL-6 levels were not normally distributed; therefore, medians and interquartile ranges were reported, and non-parametric analysis was employed. We used the Mann-Whitney U and Kruskal-Wallis tests to examine whether demographic and clinical variables are associated with h-cTnT and IL-6. Spearman correlation coefficient was used to test the correlation between h-cTnT, IL-6, and other continuous variables. This study had no missing data and a P-value of less than 0.05 was accepted as significant.

The research received approval from the Institutional Review Board (IRB) at An-Najah University, with reference number Med. June 2022/8. Patients were invited to participate in the study voluntarily, with a thorough explanation of the study's aims, objectives, and potential risks. Individuals who agreed to participate signed a written informed consent. No personally identifiable information was gathered from participants, who were referred to using unique codes to ensure confidentiality. Access to the data collected was tightly controlled and limited to the research team for the sole purpose of conducting the study.

## Results

### Background characteristics

A total of 81 HD patients were included in the study. Median age was 58 (IQR = 20–83) years, with 49 (60.5%) of these patients being male. The median duration of dialysis was 48 (IQR = 7–240) months, and 70 (86.4%) patients had fistula, whereas others had catheters. The clinical characteristics of these patients are summarized in (Table 1).

### H-cTnT and IL-6

The median values of h-cTnT and IL-6 were 62.8 ng/L (IQR = 7.6–255.6) and 2.1 pg/ml (IQR = 0.1–64.8), respectively (Table 2). Only one patient (1.2%) had h-cTnT below the upper reference limit for the average of normal population, which is it less than 14 ng/L. On the other hand, two patients (2.5%) had IL-6 above 43.5 pg/ml, the highest value detected recently in the average population, whereas the normal IL-6 range is below 7 pg/ml.

H-cTnT and IL-6 values were significantly correlated (rho = 0.463, $P<0.001$), as were h-cTnT level with age (rho = 0.519, P<0.001), albumin (rho = -0.297, p = 0.007) and transferrin saturation (rho = -0.227, P = 0.042). Furthermore, IL-6 level was correlated with age (rho = 0.422, P<0.001), albumin (rho = -0.389, P<0.001), iron (rho = -0.382, P<0.001), and transferrin saturation (rho = -0.362, P = 0.001) (Table 3).

IL-6 levels positively predicted serum concentrations of h-cTnT in simple linear regression analysis: coefficient = 1.74, standard error = 0.58, t = 2.97, P = 0.004, 95% CI = 0.57–2.91, $R2$ = 0.100 (Fig 1).

**Table 2. Distribution of Intrlukin-6 and Troponin levels among HD patients.**

|  | Median | (IQR) |
| --- | --- | --- |
| **h-cTnT level** (ng/L) | 62.8 | (7.6–255.6) |
| **IL-6 level** (pg/ml) | 2.1 | (0.1–64.8) |

h-cTnT high-sensitive cardiac troponin t, IL-6 interleukin-6.

**Table 3. Correlation between h-cTnT, IL-6, background, and lab characteristics.**

|  | hs-cTnT level | | IL-6 | |
|---|---|---|---|---|
|  | **rho** | **P- value** | **rho** | **P-value** |
| **h-cTnT**(ng/L) | 1.000 |  | 0.463 | <0.001 |
| **IL6**(pg/ml) | 0.463 | <0.001 | 1.000 |  |
| **Age** | 0.519 | <0.001 | 0.422 | <0.001 |
| **BMI** | 0.083 | 0.460 | 0.142 | 0.207 |
| **HD duration** | 0.100 | 0.376 | 0.025 | 0.822 |
| **Albumin** | -0.297 | 0.007 | -0.389 | <0.001 |
| **Ferritin** | -0.020 | 0.857 | -0.013 | 0.911 |
| **Potassium** | -0.051 | 0.654 | -0.212 | 0.057 |
| **Calcium** | 0.214 | 0.055 | 0.117 | 0.299 |
| **Iron** | -0.152 | 0.175 | -0.382 | <0.001 |
| **Phosphate** | 0.086 | 0.447 | 0.043 | 0.704 |
| **Sodium** | -0.012 | 0.917 | -0.164 | 0.144 |
| **Transferrin sat** | -0.227 | 0.042 | -0.362 | 0.001 |
| **Hemoglobin** | 0.123 | 0.273 | -0.069 | 0.542 |
| **PLT** | 0.054 | 0.634 | 0.205 | 0.067 |
| **WBC** | 0.157 | 0.162 | 0.179 | 0.111 |
| **PTH** | -0.125 | 0.267 | -0.107 | 0.340 |
| **HCO3** | 0.135 | 0.228 | 0.017 | 0.882 |

h-cTnT high-sensitive cardiac troponin t, IL-6 interleukin-6, BMI body mass index, HD hemodialysis, PLT platelet, WBC weight blood cells, PTH parathyroid hormone, HCO3 bicarbonate, rho spearman correlation coefficient.

Patients with diabetes and hypertension had significantly higher median h-cTnT levels (70.8 ng/L; P = 0.003) and (65.1ng/L; P = 0.027), respectively, as compared to non-diabetics and patients without hypertension (Fig 2 and Table 4).

Patients with a history of IHD and CVA showed significantly higher median h-cTnT levels (75.5 ng/L; P = 0.001), (70.9 ng/L; P = 0.013), respectively, than those without a history of these conditions. Moreover, males exhibited significantly higher median h-cTnT levels (73.9 ng/L; P <0.001) than females. On the other hand, the median IL-6 level did not differ significantly between groups based on gender, BMI, smoking status, and other comorbidities. In addition, there is no difference in the median values of h-cTnT and IL-6 among patients classified by vascular access type (Table 4).

## Discussion

This study demonstrated that h-cTnT and IL-6 are significantly correlated, suggesting an association between ongoing inflammation and h-cTnT concentrations in HD patients without acute symptoms of myocardial ischemia. This finding is consistent with older studies showing a correlation between h-cTnT and other inflammatory markers like CRP in HD patients [23]. On the other hand, Chen et al. showed in a cohort study of 575 dialysis patients that there was no significant association between h-cTnT and CRP level [24]. In non-dialysis patients with acute myocardial ischemia, h-cTnT and IL-6 levels are raised and used to predict further cardiac events [25,26]. Atherosclerosis is an inflammatory disease that causes extracellular matrix remodeling, oxidative stress, endothelial dysfunction, inflammation in the arterial wall, and progressive lipid accumulation [13]. It can cause multiple clinically significant CVDs, including IHD. Following myocardial ischemia, cardiomyocytes would experience irreversible

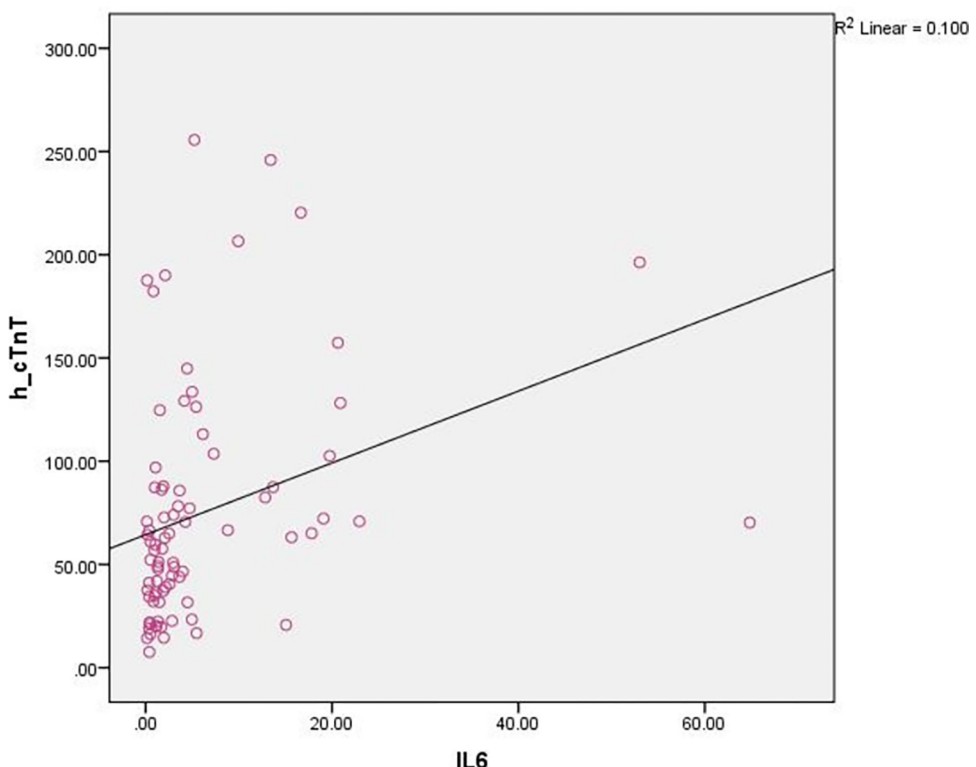

**Fig 1. Linear regression analysis of IL-6 level in relation to h-cTnT level.**

necrosis from hypoxia, followed by the infiltration of inflammatory cells and the release of pro-inflammatory factors like IL-6. Moreover, cardiomyocytes produce IL-6 under hypoxia and ischemic stress [25], which could explain the association between inflammatory markers and h-cTnT. Although this study is not powered to look at the incidence of cardiomyopathy and coronary atherosclerosis in the group of patients with elevated level of h-TnT and IL-6, particularly that those variables were found to elevated in diabetics and hypertensives who are at higher risk of having cardiomyopathy and coronary atherosclerosis [27,28]. It would be interesting to look at this possibility in future research by adding echocardiography and either invasive or non-invasive test to look for coronary atherosclerosis.

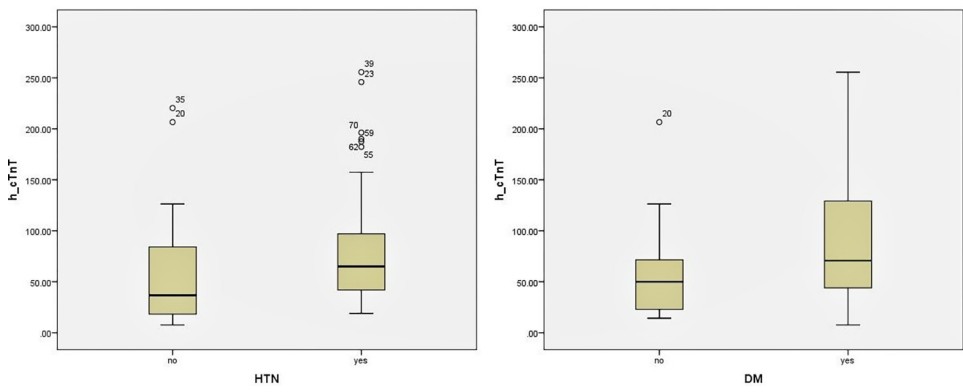

**Fig 2. h-cTnT level distribution in groups divided according to DM and HTN.**

**Table 4. Patient's background and clinical characteristics with h-cTnT and Interleukin 6.**

| | h-cTnT | | IL-6 | |
|---|---|---|---|---|
| | Median (IQR) | *P*-value* | Median (IQR) | *P*-value* |
| **Gender** | | | | |
| Male | 73.9 (19.9–255.6) | <0.001 | 2.1 (0.1–64.8) | 0.750 |
| Female | 38.9 (7.6–128.2) | | 2.3 (0.2–20.9) | |
| **BMI** | | | | |
| Underweight | 25.5 (18.9–32.0) | 0.194 | 0.6 (0.4–0.8) | 0.440 |
| Normal | 51.1 (7.6–245.9) | | 1.9 (0.2–19.7) | |
| Overweight | 70.3 (31.7–220.4) | | 3.1 (0.1–64.8) | |
| Obese | 47.9 (16.8–255.6) | | 2.1 (0.2–22.9) | |
| **Vascular Access** | | | | |
| Fistula | 63.0 (7.6–255.6) | 0.482 | 2.1 (0.1–64.8) | 0.381 |
| Central line | 41.1 (16.2–128.2) | | 2.6 (0.4–20.9) | |
| **Smoker** | | | | |
| Yes | 64.3 (7.6–255.6) | 0.467 | 1.5 (0.1–64.8) | 0.144 |
| No | 60.2 (14.3–220.4) | | 2.9 (0.2–53.0) | |
| **Diabetes** | | | | |
| Present | 70.8 (7.6–255.6) | 0.003 | 2.8 (0.1–53.0) | 0.650 |
| Absent | 50.1 (14.3–206.6) | | 1.9 (0.2–64.8) | |
| **Hypertension** | | | | |
| Present | 65.1(18.9–255.6) | 0.027 | 2.6 (0.1–64.8) | 0.393 |
| Absent | 36.8(7.6–220.4) | | 1.8 (0.2–16.7) | |
| **History of IHD** | | | | |
| Present | 75.5 (14.3–255.6) | 0.001 | 2.5 (0.1–64.8) | 0.127 |
| Absent | 46.5 (7.6–157.4) | | 1.9 (0.2–20.6) | |
| **History of CVA** | | | | |
| Present | 70.9 (20.7–196.3) | 0.013 | 4.5 (0.4–53.0) | 0.064 |
| Absent | 57.2 (7.6–255.6) | | 1.9 (0.1–64.8) | |
| **PAD** | | | | |
| Present | 57.6 (14.3–245.9) | 0.971 | 1.9 (0.2–22.9) | 0.601 |
| Absent | 63.0 (7.6–255.6) | | 2.1 (0.1–64.8) | |
| **Duration of dialysis** | | | | |
| < 18 months | 43.0 (18.9–206.6) | 0.625 | 3.5 (0.2–53.0) | 0.546 |
| ≥ 18 months | 63.7 (7.6–255.6) | | 2.0 (0.1–64.8) | |

IQR interquartile range, BMI body mass index, IHD ischemic heart disease, CVA cerebrovascular accident, PDA peripheral arterial disease. * Mann-Whinty U test and Kruskal-Wallis test.

The correlations with other negative acute phase reactants like transferrin saturation and serum albumin further underscored the association between h-cTnT and inflammation in the current study. As expected, we also found that IL6, a potent inflammatory marker, has an inverse correlation with serum iron level, transferrin saturation, and albumin level. Inflammation affects intestinal iron absorption and sequester iron in macrophages, decreasing serum iron levels and transferrin saturation [29]. Ferritin is also a marker of acute and chronic inflammation. Patients with CKD had hyperferritinemia due to iron overload and chronic inflammation [30].

In contrast to a previous study that showed a positive correlation between ferritin and inflammatory markers like IL-6 and CRP in patients with HD [31], we found that ferritin has an inverse correlation with h-cTnT and IL-6. However, our study's results were not statistically

significant, which could be attributed to the small sample size. Malnutrition may cause a drop in serum albumin level, although inflammation can cause this as albumin is a negative acute phase reactant [32]. Thus, HD patients with the lowest albumin concentrations are those with both vascular disease and inflammation. In addition, previous studies demonstrated that hypoalbuminemia is one of the strongest predictors of mortality in ESRD [33]. As we discovered a strong association between inflammatory markers and h-cTnT, the present question is whether the use of drugs with anti-inflammatory effects will help in decreasing CVD mortality in asymptomatic HD patients.

The minimum and maximum levels of h-cTnT in HD patients without acute coronary syndrome were also assessed in this study; the h-cTnT range was 7.6 ng/L to 255.6 ng/L. Even higher h-cTnT level (738.30 ng/L) has been reported in HD patients without acute ischemia [34]. As was previously mentioned, even in the absence of acute episodes of myocardial ischemia, blood troponin concentrations in ESRD patients tend to be greater than in the general population [35]. We also discovered many patients with elevated h-cTnT (98.8%), similar to other studies on h-cTnT [36,37]. Although the pathophysiological basis and the biochemical processes underlying the elevated h-cTnT concentrations in ESRD are still unclear, it is almost attributed to chronic myocardial damage rather than acute myocardial infarction as these patients were clinically stable and did not have any signs or symptoms of acute ischemia [34]. Estimated glomerular filtration rates affect the metabolism and clearance of h-cTnT, so this could explain the elevation of troponin in CKD [38]. Volume removal and overload may play a role in inducing ischemia and elevation in troponin. Rapid changes in blood pressure during dialysis treatment are considered a risk factor for IHD. Intradialytic ultrafiltration causes hypotension and a decrease in venous return with a subsequent increase in the stretching of the heart without compensation, particularly in patients with chronic dialysis and heart failure, which leads to myocardial ischemia due to hypoperfusion [39], thus higher troponin levels.

Several hypotheses have been proposed about the elevation of cardiac troponins in HD patients that suggest myocardial injury. Left ventricular hypertrophy was a factor highly predictive of high serum levels of cardiac markers [40]. Some cross-sectional studies on clinically stable patients with ESRD have found that hs-cTnI was strongly correlated to left ventricular dysfunction and h-cTnT was correlated to coronary artery disease based on a single troponin value [41]. Rather than coronary artery stenosis, microvascular lesions or direct injury to myocardial cells (e.g., toxic, stretching, hypoxia, apoptosis) encountered in the cardiomyopathy of HD patients was associated with high cardiac troponins. The co-existence of heart failure, even without acute ischemia, non-ischemic clinical conditions, including tachy/bradyarrhythmias and aortic valve disease, were all associated with high cardiac troponins [40]. Another study found that high cardiac troponin level was associated with increased risk for all-cause and CVD mortality, as mortality risk was doubled in patients with persistently high levels of h-cTnT in ESRD [42]. However, what remains to be established is whether a higher reference limit of cardiac troponins will be used to diagnose acute coronary syndrome in patients with ESRD and if these patients with acute elevations in troponins are accurately identified as those who would benefit from early coronary interventions, such as administration of IIb/IIIa anti-platelet inhibitors or coronary revascularization, as was confirmed in the general population. Further studies are needed to investigate whether more specific diagnostic tests, such as coronary angiography, would lead to interventions that improve the outcomes in patients with chronic elevations of troponins and inflammatory markers.

Diabetes, hypertension, previous history of coronary artery disease, and cerebrovascular accident were found in this study to be associated with elevated concentrations of h-cTnT, which have been reported by other studies [24,43,44]. Findings of higher serum h-cTnT in diabetic HD patients are consistent with previous data from HD patients reported by others [36].

Advanced glycosylation end-products could alter membrane integrity and promote h-cTnT expression in noncardiac cells, which could account for this elevation. Chronic hyperglycemia also affects heart microcirculation, which results in microvascular damage and, in turn, ischemia, both of which raise troponin levels [45]. Šimić et al. suggested using other cardiac damage markers, such as copeptin and heart fatty acid binding protein, which have different elimination mechanisms [45]. Moreover, diabetic patients are more likely to develop microangiopathy and myocardial ischemia, which may be clinically silent and undiagnosed in these patients. We also found a significant association between h-cTnT level and hypertension in the current investigation, as chronic hypertension is associated with arterial wall stiffness and atherosclerosis.

Another key finding in our study is that age and male gender correlated significantly with the h-cTnT level of the HD population, supporting earlier reports [35]. This study discovered that age and the male gender are both independently related to the risk of increasing h-cTnT levels. While it seems that local inflammation, microvascular damage, and atherosclerosis are connected with older age, the explanation of male gender is less obvious. Male gender is associated with a higher left ventricular mass index [46], but in another study, these associations persisted after adjustment for the left ventricular mass index [35]. In our study, neither the duration of HD nor the type of vascular access was associated with h-cTnT levels.

This study has possible limitations. First, the cross-sectional study design limits the ability to suggest causality between related variables. Second, the results should be cautiously generalized because the study was conducted at a single dialysis unit with a small sample size. Thirdly, not all patients were evaluated with echocardiography, so elevated cardiac troponin could be associated with other causes like left ventricular hypertrophy rather than atherosclerosis. Since cardiac catheterization and ECG were not used to investigate patients with high cardiac troponin levels, it is impossible to rule out the possibility of an acute cardiac event in these asymptomatic patients.

## Conclusion

The findings suggest that elevated h-cTnT in asymptomatic HD patients may be linked to chronic myocardial damage rather than acute myocardial infarction, as these patients were clinically stable and lacked symptoms of acute ischemia. In addition to IL-6, several clinical factors, such as diabetes, hypertension, a previous history of coronary artery disease, and cerebrovascular accidents, were associated with elevated h-cTnT levels, highlighting the importance of identifying high-risk patients. Early detection of cardiovascular disease in asymptomatic HD patients is crucial for preventing adverse outcomes and reducing mortality. This study raises the possibility of using anti-inflammatory interventions to reduce this population's CVD risk. Further research is needed to explore whether more specific diagnostic tests and interventions can improve outcomes in patients with chronic elevations of troponins and inflammatory markers.

## Supporting information

**S1 File.**
(PDF)

## Acknowledgments

The authors thank the patients who participated in this study. We would also like to thank the dialysis unit staff at An-Najah National University Hospital for their assistance and facilitation of data collection.

## Author Contributions

**Conceptualization:** Yahya Ismael, Zakaria Hamdan, Zaher Nazzal.

**Data curation:** Leen Ibrahim, Katreen Yasin, Leen Abbas, Ahmed Mousa, Mohammad Alkarajeh, Zaher Nazzal.

**Formal analysis:** Katreen Yasin, Leen Abbas, Mohammad Alkarajeh, Zaher Nazzal.

**Investigation:** Leen Ibrahim, Ahmed Mousa.

**Methodology:** Yahya Ismael, Ahmed Mousa, Mohammad Alkarajeh, Zakaria Hamdan.

**Project administration:** Zaher Nazzal.

**Software:** Leen Ibrahim, Katreen Yasin, Leen Abbas, Mohammad Alkarajeh.

**Supervision:** Yahya Ismael, Zakaria Hamdan, Zaher Nazzal.

**Writing – original draft:** Leen Ibrahim, Katreen Yasin, Leen Abbas, Mohammad Alkarajeh, Zaher Nazzal.

**Writing – review & editing:** Yahya Ismael, Ahmed Mousa, Zakaria Hamdan, Zaher Nazzal.

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
