## [Decision Letter · Decision Letter 0]

12 Dec 2023

PONE-D-23-33372Exploring the relation between Interleukin-6 and high-sensitive cardiac troponin T in a symptomatic hemodialysis patient: A cross-sectional study.PLOS ONE

Dear Dr. Nazzal,

Thank you for submitting your manuscript to PLOS ONE. After careful consideration, we feel that it has merit but does not fully meet PLOS ONE’s publication criteria as it currently stands. Therefore, we invite you to submit a revised version of the manuscript that addresses the points raised during the review process.

ACADEMIC EDITOR: Dear authors, please address the concerns raised by Reviewers.

We look forward to receiving your revised manuscript.

Kind regards,

Yashendra Sethi

Academic Editor

PLOS ONE

Journal Requirements:

Additional Editor Comments:

We have now received comments from reviewers and they have raised some minor concerns. Please address them before we can proceed further.

Reviewers' comments:

Reviewer's Responses to Questions

Comments to the Author

1. Is the manuscript technically sound, and do the data support the conclusions?

Reviewer #1: Yes

Reviewer #2: Partly

2. Has the statistical analysis been performed appropriately and rigorously? 

Reviewer #1: Yes

Reviewer #2: Yes

3. Have the authors made all data underlying the findings in their manuscript fully available?

Reviewer #1: Yes

Reviewer #2: Yes

4. Is the manuscript presented in an intelligible fashion and written in standard English?

Reviewer #1: Yes

Reviewer #2: Yes

5. Review Comments to the Author

Reviewer #1: It is interesting in terms of the metadology and subject matter of the study, but as mentioned in the limitations section

265 was performed in a single dialysis unit with a small sample size. Not all patients were evaluated by echocardiography, so high cardiac troponin values may be related to left cardiac function. Since cardiac catheterization was not used, it is not possible to exclude coronary artery disease.

But because of the nature of the subject, I think it can be published.

Reviewer #2: I would like to appreciate the efforts by the authors for this study, considering that cardiovascular disease is among the top causes of mortality in those suffering from CKD. However, I have few queries and suggestions regarding the study which I have mentioned below.

1.In the heading it is mentioned ‘ a symptomatic’. Please correct this to asymptomatic

2.Please mention the effect of chronic kidney disease and hemodialysis on levels of Troponin and Il 6. Cardiac troponins are known to be chronically elevated in CKD cases.

3.Please mention the normal values of Troponin and Il 6?

4.You have mentioned that hemodialysis has further detrimental effects on cardiovascular system secondary to increased oxidative stress.It would be better if you could cite data regarding levels of troponin and Il 6 among CKD cases on hemodialysis compared to CKD cases not on hemodialysis to emphasise the impact of hemodialysis on inflammation to the readers.

5.Were those with underlying infection excluded as it is known to cause elevation in levels of Il 6

6. PLOS authors have the option to publish the peer review history of their article (what does this mean?). If published, this will include your full peer review and any attached files.

Do you want your identity to be public for this peer review? For information about this choice, including consent withdrawal, please see our Privacy Policy.

Reviewer #1: No

Reviewer #2: No

---

## [Author Response · Author response to Decision Letter 0]

14 Dec 2023

Reviewer 1 comments:

Reviewer #1: It is interesting in terms of the methodology and subject matter of the study, but as mentioned in the limitations section

265 was performed in a single dialysis unit with a small sample size. Not all patients were evaluated by echocardiography, so high cardiac troponin values may be related to left cardiac function. Since cardiac catheterization was not used, it is not possible to exclude coronary artery disease.

But because of the nature of the subject, I think it can be published.

Authors’ Response: I appreciate your supportive words, and we agree with your concerns. We added a paragraph in the discussion supporting your idea.

Reviewer 2 comments

Reviewer #2: I would like to appreciate the efforts by the authors for this study, considering that cardiovascular disease is among the top causes of mortality in those suffering from CKD. However, I have few queries and suggestions regarding the study which I have mentioned below.

Authors’ Response: Thank you for your kind and encouraging words

1.In the heading it is mentioned ‘ a symptomatic’. Please correct this to asymptomatic

Authors’ Response: Thank you for noting this. We edited it

2.Please mention the effect of chronic kidney disease and hemodialysis on levels of Troponin and Il 6. Cardiac troponins are known to be chronically elevated in CKD cases.

Authors’ Response: We value your suggestion and agree with it. We have incorporated this into both the introduction and discussion sections, supported by relevant references. Many studies have highlighted the increase of IL-6 and troponin in individuals undergoing hemodialysis and those with chronic kidney disease (CKD).

3.Please mention the normal values of Troponin and Il 6?

Authors’ Response: We acknowledge your comment. The normal troponin level is under 14 ng/L, and the normal IL-6 level is under 7 pg/ml. We have integrated this information into the relevant sections of the methods and results.

4.You have mentioned that hemodialysis has further detrimental effects on cardiovascular system secondary to increased oxidative stress. It would be better if you could cite data regarding levels of troponin and Il 6 among CKD cases on hemodialysis compared to CKD cases not on hemodialysis to emphasize the impact of hemodialysis on inflammation to the readers.

Authors’ Response: Thank you for suggesting this. We cited data regarding your point in the introduction session.

5.Were those with underlying infection excluded as it is known to cause elevation in levels of Il 6.

Authors’ Response: Thank you for bringing this to our attention. We missed including this factor in the exclusion criteria, but we have applied it and made certain that no participants in the study had any had an underlying infection during the study conduction. We added it in the eligibility criteria section

---

## [Editor Report · Decision Letter 1]

27 Dec 2023

Full title: Exploring the relation between Interleukin-6 and high-sensitive cardiac troponin T in asymptomatic hemodialysis patient: A cross-sectional study.

PONE-D-23-33372R1

Dear Dr. Nazzal,

We’re pleased to inform you that your manuscript has been judged scientifically suitable for publication and will be formally accepted for publication once it meets all outstanding technical requirements.

Kind regards,

Yashendra Sethi

Academic Editor

PLOS ONE

Additional Editor Comments (optional):

Congratulations on addressing all the comments and reaching a good revised version. We can now accept this for publication.

Thank you for your contribution.

Merry Christmas and a happy new year!

Hope this acceptance starts your year with smiles.

Thank you for your contribution!
---

## [Editor Report · Acceptance letter]

17 Jan 2024

PONE-D-23-33372R1 

PLOS ONE

Dear Dr. Nazzal, 

I'm pleased to inform you that your manuscript has been deemed suitable for publication in PLOS ONE. Congratulations! Your manuscript is now being handed over to our production team.

Kind regards, 

on behalf of

Dr. Yashendra Sethi 

Academic Editor

PLOS ONE